# A multi-layer mean-field model of the cerebellum embedding microstructure and population-specific dynamics

Roberta Maria Lorenzi[1]*, Alice Geminiani[1], Yann Zerlaut[2], Marialaura De Grazia[1], Alain Destexhe[3], Claudia A. M. Gandini Wheeler-Kingshott[1,4,5], Fulvia Palesi[1], Claudia Casellato[1], Egidio D'Angelo[1,5]

1 Department of Brain and Behavioural Sciences, University of Pavia, Pavia, Italy, 2 Institut du Cerveau—Paris Brain Institute—ICM, Inserm, CNRS, APHP, Hôpital de la Pitié Salpêtrière, Paris, France, 3 Paris-Saclay University, CNRS, Saclay, France, 4 NMR Research Unit, Queen Square Multiple Sclerosis Centre, Department of Neuroinflammation, UCL Queen Square Institute of Neurology, UCL, London, United Kingdom, 5 Brain Connectivity Center, IRCCS Mondino Foundation, Pavia, Italy

* robertamaria.lorenzi01@universitadipavia.it

**Data Availability Statement:** The codes will be available at https://github.com/RobertaMLo/CRBL_MF.

## Abstract

Mean-field (MF) models are computational formalism used to summarize in a few statistical parameters the salient biophysical properties of an inter-wired neuronal network. Their formalism normally incorporates different types of neurons and synapses along with their topological organization. MFs are crucial to efficiently implement the computational modules of large-scale models of brain function, maintaining the specificity of local cortical microcircuits. While MFs have been generated for the isocortex, they are still missing for other parts of the brain. Here we have designed and simulated a multi-layer MF of the cerebellar microcircuit (including Granule Cells, Golgi Cells, Molecular Layer Interneurons, and Purkinje Cells) and validated it against experimental data and the corresponding spiking neural network (SNN) microcircuit model. The cerebellar MF was built using a system of equations, where properties of neuronal populations and topological parameters are embedded in inter-dependent transfer functions. The model time constant was optimised using local field potentials recorded experimentally from acute mouse cerebellar slices as a template. The MF reproduced the average dynamics of different neuronal populations in response to various input patterns and predicted the modulation of the Purkinje Cells firing depending on cortical plasticity, which drives learning in associative tasks, and the level of feedforward inhibition. The cerebellar MF provides a computationally efficient tool for future investigations of the causal relationship between microscopic neuronal properties and ensemble brain activity in virtual brain models addressing both physiological and pathological conditions.

## Author summary

Whole-brain dynamics can be simulated using cortical and subcortical mean-field models, which provide a population-level description of the underlying neuronal dynamics. While mean-field models of the isocortex have recently been developed, a mean-field model of

**Funding:** This research has received funding from the European Union's Horizon 2020 Framework Program for Research and Innovation under the Specific Grant Agreement No. 945539 (Human Brain Project SGA3) to ED, CGWK, FP and AD, and under the Marie Sklodowska-Curie grant agreement No. 892175 to YZ. CGWK received funding from BRC (#BRC704/CAP/CGW), MRC (#MR/S026088/1), Ataxia UK, MS Society (#77), Wings for Life (#169111). This research has also received funding from Centro Fermi project "Local Neuronal Microcircuits" to ED. Special acknowledgement to EBRAINS and FENIX for informatic support and infrastructure. RL, and AG have been supported by Human Brain Project SGA3. This work was also supported by #NEXTGENERATIONEU (NGEU) and funded by the Ministry of University and Research (MUR), National Recovery and Resilience Plan (NRRP), project MNESYS (PE0000006) – A Multiscale integrated approach to the study of the nervous system in health and disease (DN. 1553 11.10.2022) to ED, CGWK, and CC, and Project EBRAINS-Italy (IR00011) - (M4C2 Line 3.1 of the PNRR, Action 3.1.1 - CUP B51E22000150006) to ED and CC. The funders had no role in study design, data collection and analysis, decision to publish, or preparation of the manuscript.

**Competing interests:** I have read the journal's policy and the authors of this manuscript have the following competing interests: CGWK is a shareholder in Queen Square Analytics Ltd.

the cerebellar cortex is still missing but is much needed given its specific structural and functional organization. Thus, we developed the first biologically grounded mean-field model of the cerebellar cortex, which embeds a realistic network architecture with 4 main neuron populations (granule cells, Golgi cells, Purkinje cells, molecular layer interneurons) represented with non-linear neuronal models embedding a set of neuron- and synapse-specific parameters. The model was validated and tuned against experimental data and spiking neural network simulations. The mean-field model can reproduce local neural dynamics elicited by different cortical inputs and accurately predicts population-specific activity patterns. The possibility of tuning multiple neuronal and synaptic parameters allows to capture local neural dynamics both in physiological and pathological conditions. The cerebellar mean-field model is now ready to be integrated into brain dynamic simulators, fostering a deeper understanding of the cerebellar impact on brain dynamics in functional and dysfunctional states.

## 1. Introduction

Realistic modelling of brain function is opening new frontiers for experimental and clinical research towards personalized and precision medicine [1,2]. Brain models can be developed at different scales, ranging from microscopic properties of neurons and microcircuits to the ensemble behavior of the whole brain. Arguably, a model able to reproduce realistic behavior of specific neuronal populations across scales would increase the fidelity of modelling single brain regions and consequently improving the accuracy of whole-brain dynamic simulations [3]. This approach, though, bears conceptual and practical drawbacks. At the microscale level, e.g. at the $\mu m^3$ scale, Spiking Neural Networks (SNNs) reproduce neural circuits as a set of interconnected neurons [4–6]: the state of each neuron and synapse in the network is updated at each simulation step, allowing the investigation of neural circuits functioning with a high level of granularity and biological plausibility. However, this degree of detail is hard to manage when simulating brain signals, like those derived from electroencephalography (EEG) or functional magnetic resonance imaging (fMRI), representing the activity of large cortical regions, e.g., at the $mm^3$ scale. To manage the high complexity of brain signals, the dynamics of neuronal populations have been condensed into ensemble density models called neural masses. These provide a description of the expected values of neuronal activity states, under the assumption that the equilibrium density has a point mass [7,8]. Neural fields are obtained from neural mass models when considering spatial information: these can be used to model spatial propagation of activity throughout brain volumes [9]. Despite being computationally efficient and easy to fit on brain signal data, neural mass and neural field models lack a direct link to the real behavior of neurons at the microscopic scale; this limits their applicability to investigate the neuronal bases of brain dynamics and the causal relationships between neural mechanisms at different scales. The mean-field (MF) approximation theory describes collective neural activities maintaining a direct link with the corresponding spiking neural network. The MF model formalism approximates high-dimensional random systems by averaging their original properties over degrees of freedom to compute the first two statistical moments of the system variables (e.g., mean and variance of the population firing rates in a neural network). Different mathematical approaches exist to derive such statistical descriptions depending on the complexity of the underlying network. While theoretical derivations in simpler settings allow to capture firing dynamics up to fast synchronized events (e.g., Lorentzian [10,11] or finite-size effect-based formalisms [12,13]), the network

considered here impedes such an approach. Instead, we use a formalism based on the definition of a transfer function (TF) that models the neuronal input-output relationship to investigate the properties of network responses to different input [7,14,15]. The MF developed here is based on the TF approach that summarizes the neuronal and connectivity properties of an entire SNN through *ad-hoc* transfer functions [15–17] and captures the statistical properties of network activity by computing the probabilistic evolution of neuronal states at subsequent time intervals [14,18,19].

MFs can be used to investigate macroscale phenomena, such as brain rhythms and coherent oscillations [20], and are computationally advantageous, with increased computational speed and low memory requirements compared to SNNs. Among current limitations, this MFs do not capture in full the complex properties of specific neuronal populations and are valid only in certain firing regimes, e.g., at low frequency [21].

Moreover, while a diversification of MFs for specific cortical regions has been proposed [22–28], the development of MFs for subcortical regions has seen only a few attempts [29–32], despite the fundamental role such regions exert in controlling brain dynamics and behavior [33–38]. In particular, the cerebellum has a dense connectivity with the thalamus and the cerebral cortex and impacts remarkably on whole-brain dynamics, as shown in resting-state and task-dependent fMRI [36,39,40]. The cerebellum highly specific cortical microcircuit structure and its neuronal population specific dynamics are prompting for the development of specific MFs to be included into whole-brain simulators [41].

The cerebellar cortex, indeed, receives inputs from mossy fibers and climbing fibers and sends outputs to the deep cerebellar nuclei. Granule Cells (GrC), Golgi Cells (GoC), Molecular Layer Interneuron (MLI) and Purkinje Cells (PC) constitute the backbone of the cerebellar cortex, which shows a unique anisotropic geometry supporting a forward architecture with limited lateral connectivity, recurrent excitation and inhibition, and an inhibitory output projecting from PC to the Deep Cerebellar Nuclei. These properties, along with the above-mentioned neuronal types, differ remarkably from those of the cerebral cortex, which is typically modelled using MFs including one excitatory and one inhibitory population that are reciprocally connected, have recurrent excitation and project an excitatory output. This general MF design neglects specific properties of the cerebellum such as the multilayer organization and the lack of recurrent excitation, justifying the need of a specific cerebellar MF model with appropriate circuitry and temporal dynamics. All the types of neurons of the cerebellar cortex have been characterized through electrophysiology experiments in rodents *in vitro* and *in vivo* [42–46] and have been, subsequently, represented by detailed multicompartmental models [47–51]; moreover, these detailed models have been simplified into point-neuron models [52–54], and embedded in network models of the cerebellar microcircuit [44,52,55–57]. Thus, the cerebellum provides an ideal substrate for generating a MF, with internal dynamics that can be remapped onto a precise physiological counterpart. Finally, the MF model of the cerebellum can be validated against the existing rich electrophysiological data and detailed cerebellar microcircuit activity simulations. In this work we have developed and validated a multi-layer MF of the cerebellar cortex, which maintains the salient properties of the inter-wired cerebellar neuronal populations. Indeed, the MF was derived from a biology-grounded model of the cerebellar microcircuit, which was used to define the topology and tune the parameters of the MF, and it was then validated against a rich set of SNN outputs. The future prospective is to integrate the present mesoscopic cerebellar MF as a module of macroscale models, e.g. Dynamic Causal Modeling (DCM) [58–60] or The Virtual Brain (TVB) [41], to simulate the cerebellar contribution to brain activity in physiological and pathological conditions.

## 2. Methods

In this section we describe the development, tuning, validation, and application of a multi-layer MF of the cerebellar circuit (Fig 1). The MF formalism provides a statistical summary of a SNN activity through the first two statistical moments (i.e. average and variance) of the population firing rates [14]. Here the SNN bottom-up modelling approach is merged with the standard MF mathematical formalism to obtain a multi-layer MF of the cerebellar cortex. In order, we present the cerebellar SNN model used as the structural and functional reference of the MF (**section 2.1**), the design of the MF architecture based on cerebellar topology (**section 2.2.1 and 2.2.3**), the implementation of the MF equations derived with an heuristic approach (**section 2.2.2**) [19,21,61] the protocols used to optimise the MF time constant (**section 2.2.4**) and to validate the MF (**section 2.3**), the applications of the MF to predict the activity modulation induced by different levels of synaptic plasticity and of inhibitory control (**section 2.4**). For the sake of simplicity, activity patterns in the MF are described using a terminology derived from the SNN, so that a rate-coded waveform with a peak followed by a reduction is called burst-pause to suggest that the generative mechanism is the same as that of a spike-coded pattern.

### 2.1 SNN model

This cerebellar cortex model was built using the Brain Scaffold Builder (BSB) (https://bsb.readthedocs.io/en/latest/), a neuroinformatic framework allowing a detailed microcircuit

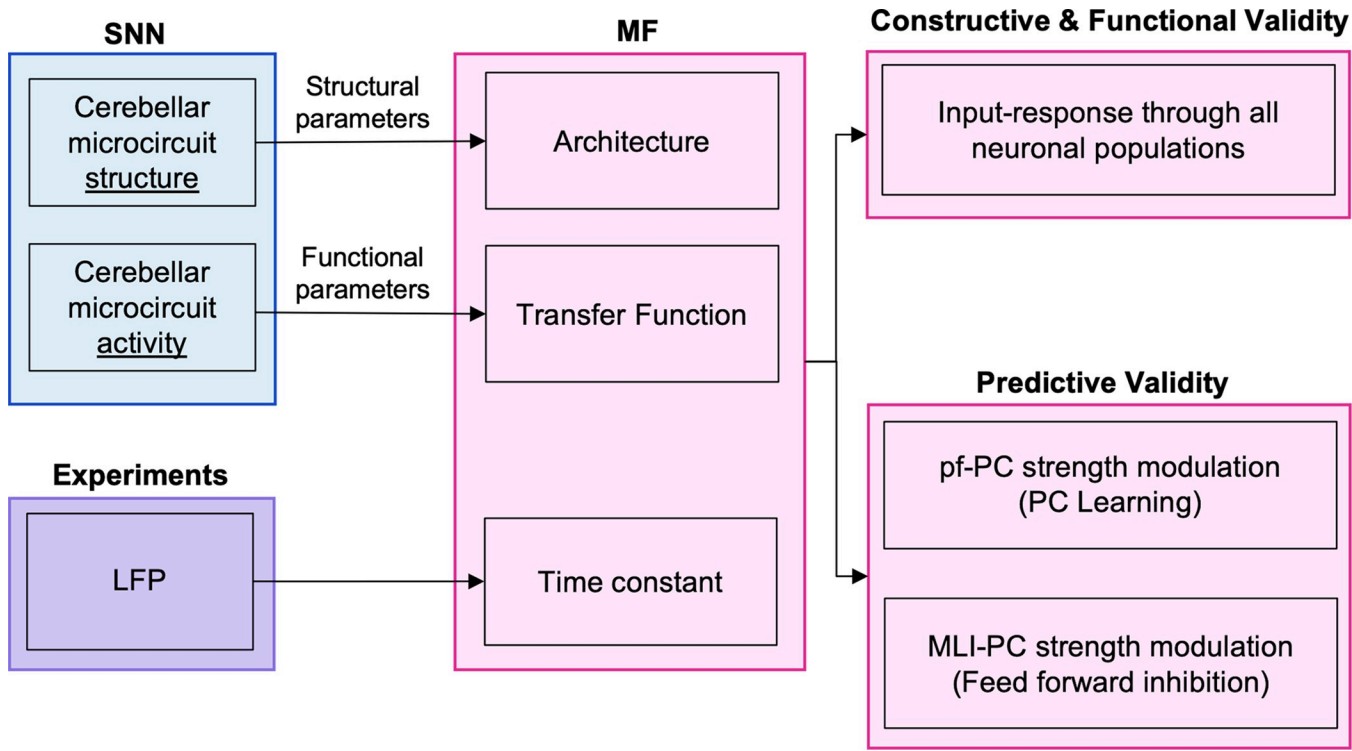

**Fig 1. Pipeline of the multi-layer cerebellar Mean Field (MF) model.** The workflow of the study is represented. MF was designed based on structural and functional parameters extracted from Spiking Neural Network (SNN) simulations. The time constant of the resulting MF was optimized against Local Field Potential (LFP) experimental data. The model was first validated against neural activity of SNN with different stimulation protocols (Constructive & Functional validity) to compare MF simulations with SNN outcomes. MF was also tested to reproduce the effect of synaptic plasticity (Predictive Validity) in Purkinje Cells (PC) by modulating the parallel fibers (pf) and molecular layer interneurons (MLI) synaptic strength to investigate respectively learning mechanisms and feed forward inhibition.

reconstruction based on neuron morphologies and orientations and the incorporation of active neuronal and synaptic properties [56]. The model was simulated as a SNN with ~$3 \times 10^4$ extended-Generalised Leaky Integrate and Fire (E-GLIF) neurons [52,54] and ~$1.5 \times 10^6$ alpha-shaped conductance-based synapses [62]. The SNN simulations were performed using NEST version 2.18 (https://zenodo.org/record/2605422) [4,63].

**2.1.1 Neuron model.** The E-GLIF formalism describes the time evolution of membrane potential ($V_m$) depending on two intrinsic currents to generate slow adaptation ($I_{adap}$) and fast depolarisation ($I_{dep}$), using a the system of three Ordinary Differential Equations [54]

$$\begin{cases} \dfrac{dV_m(t)}{dt} = \dfrac{1}{C_m}\left(\dfrac{C_m}{\tau_m}(V_m(t) - E_{rev}) - I_{adap}(t) + I_{dep}(t) + I_e + I_{syn}\right) \\[2ex] \dfrac{dI_{adap}(t)}{dt} = k_{adap}(V_m(t) - E_{rev}) - k_2 I_{dep}(t) \\[2ex] \dfrac{dI_{dep}(t)}{dt} = k_1 I_{dep}(t) \end{cases} \tag{1}$$

where $I_{syn}$ = synaptic current (it models the synaptic stimulus, see section 2.1.2); $C_m$ = membrane capacitance; $\tau_m$ = membrane time constant; $E_{rev}$ = reversal potential; $I_e$ = endogenous current; $k_{adap}$ and $k_2$ = adaptation constants; $k_1$ = decay rate of $I_{dep}$. When a spike occurs, state variables are updated as follows:

$$\begin{aligned} V_m(t_{spk}^+) &= V_r \\ I_{adap}(t_{spk}^+) &= I_{adap} t_{spk}^+ + A_2 \\ I_{dep}(t_{spk}^+) &= A_1 \end{aligned} \tag{2}$$

where $t_{spk}^+$ = time instant immediately following the spike time $t_{spk}$; $V_r$ = reset potential; $A_2$, $A_1$ = model currents update constants. E-GLIF models were implemented using sets of parameters specific for each neuronal population [52] as shown in S1 Table.

**2.1.2 Synaptic model.** Connections between neural populations were modelled as conductance-based synapses:

$$I_{syn}(t) = g_{syn}(t)(V_m(t) - E_{rev}) \tag{3}$$

When a spike occurs, the conductance $g_{syn}$ changes according to an alpha function:

$$g_{syn}(t) = G_{syn}\frac{t - t_{spk}}{\tau_{syn}}e^{1-\frac{t-t_{spk}}{\tau_{syn}}} \tag{4}$$

where $G_{syn}$ is the maximum conductance change and $\tau_{syn}$ the synaptic time constant. E-GLIF neuron models and conductance-based synaptic models used in SNN simulations provided the functional reference of cerebellar spiking activity for MF development.

## 2.2 Mean Field (MF) design

The design of the cerebellar multi-layer MF was based on the extensive knowledge of cerebellar anatomy and physiology summarized in previous cerebellar cortex network models [52,54,56].

**2.2.1 Architecture.** The cerebellar MF included the main neuronal populations of the cerebellar cortex—GrC, GoC, MLI and PC (Fig 2) and the corresponding excitatory and inhibitory synapses. The granular layer at the cerebellar input stage includes GrC and GoC receiving the external input ($\nu_{drive}$) from mossy fibers. GrC excite GoC, which, in turn, inhibit themselves and GrC forming recurrent loops. GrC represent the excitatory input for the molecular

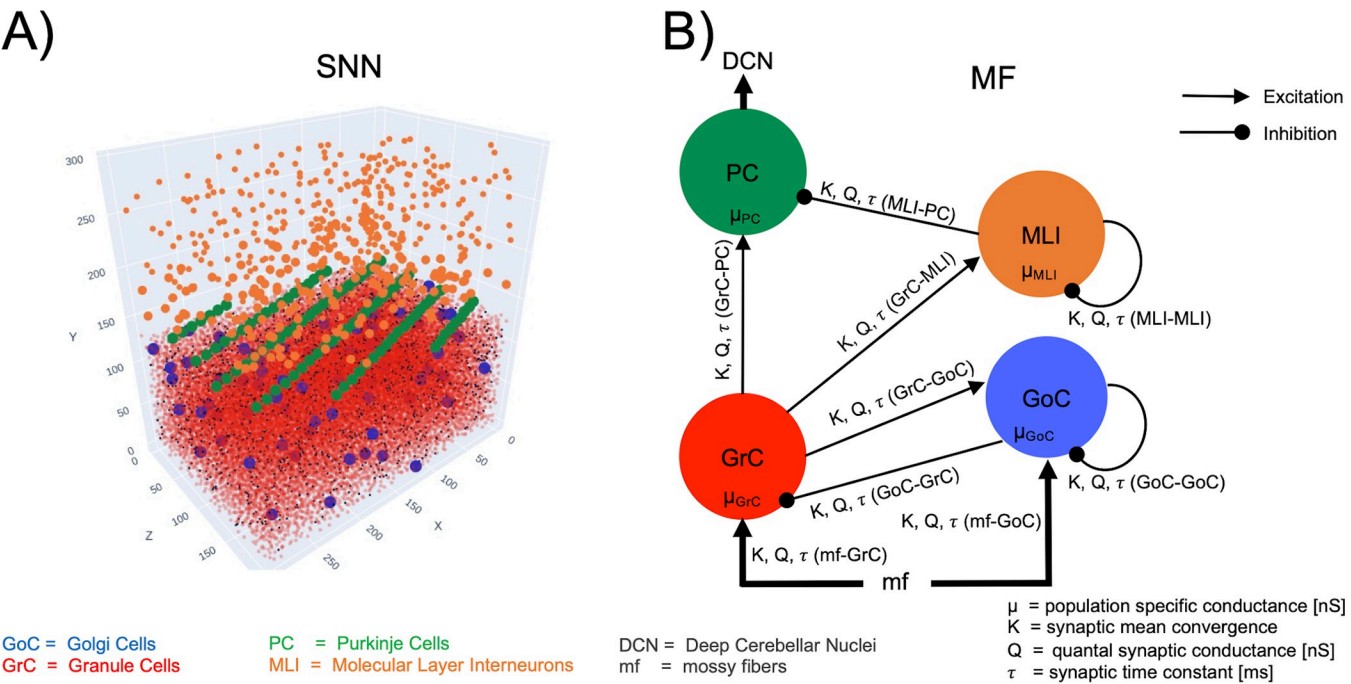

**Fig 2. Multi-layer mean-field architecture and parameters. A**: Spiking Neural Network model (SNN). The cerebellar cortical volume (length x width x height = 300 x = 200 x295 μm3) contained a total of 29230 neurons including 28615 Granule Cells (GrC), 70 Golgi Cells (GoC), 446 Molecular Layer Interneurons (MLI)and 99 Purkinje Cells (PC). **B**: Multi-layer MF architecture with neuronal populations connected according to anatomical knowledge. The main cerebellar neuron types are included: GrC and GoC, receiving input from mossy fibers (mf), MLI, and PC which are the sole output of the cerebellar cortex sending their projections to the Deep Cerebellar Nuclei (DCN). Each population receives excitatory and/or inhibitory input activity) from presynaptic populations, depending on their specific conductance μ and on the synaptic properties of each connection (K = synaptic mean convergence, Q = synaptic conductance, and τ = synaptic time constant).

layer constituted by MLI and PC. MLI inhibit PC, which are the sole output of the cerebellar cortex and shape the deep cerebellar nuclei activity through inhibition. Although other neurons have been reported to play a role in the cerebellar microcircuit (e.g. Lugaro cells [64], and unipolar brush cells [65]), for the sake of simplicity we have limited the present model to the canonical architecture that is thought to generate the core network computations.

To connect the nodes of the MF network, synaptic parameters were set according to those of the reference SNN (S2 Table). The connection probability of each connection type (K) was derived from the synaptic convergence on the postsynaptic neuronal population in a cerebellar cortical volume [56]. The quantal synaptic conductance and synaptic time decay (Q, τ) were derived from the weights and time constants of the corresponding synapse models [52].

**2.2.2 Transfer Function (TF) Computation.** The TF is defined for each population as a mathematical construct that takes the activity of the presynaptic population ($v_s$) as input and provides an average population activity signal as output ($v_{out}$) [18].

A purely analytic derivation of the TF (using approximations and stochastic calculus, e.g. [15]) was not possible given the complexity of the neuronal (E-GLIF) and synaptic (alpha-waveform) models considered here. Therefore, TFs, here, relied on a semi-analytical approach that couples an approximate analytical estimate with an optimization step to capture the firing response of analytically intractable models [19], see also [16] for a similar approach). More details can be found in [19,61] but we summarize the approach below.

The analytical template for the TF (indicated with F in the equations for sake of simplicity) of all neuron types is derived from the probability to be above threshold in the fluctuation-

driven regime [18]:

$$v_{out} = F_p(v_s) = \alpha \frac{1}{2\tau_V} erfc\left(\frac{V_{thre}^{eff} - \mu_V}{\sqrt{2}\sigma_V}\right) \quad (5)$$

where the complementary *erfc* is the error function while $\mu_V$, $\sigma_V^2$ and $\tau_V$ are the average, variance and autocorrelation time respectively of the membrane potential fluctuations. Two phenomenological terms were introduced: $V_{thre}^{eff}$, an effective firing threshold to capture the impact of single cell non-linearities on firing response [19] and $\alpha$, a multiplicative factor to adapt the equations also to high input frequency regimes [21]. Those two terms were optimized for each neuron type (see steps *d* and *e*, below) from single neuron simulations of input-output transformation in terms of firing rate (i.e., the numerical TF, see step *b*). The TF depends on the statistical properties of the subthreshold membrane voltage dynamics (mean = $\mu_V$, standard deviation = $\sigma_V^2$ and autocorrelation time $\tau_V$, calculated in step *c*). These in turns depend on the average population conductances that are computed with the biologically-grounded functional parameters derived from SNN models at single neuron resolution (step *a*), bringing the physiological properties into the MF mathematical construct.

  *a) Equations of Population-specific conductance*. For each neuronal population, the average conductance was defined as a function of the presynaptic inputs, according to the topology described in **section 2.1** (Fig 3.):

$$\mu_p = \sum_s K_{s-p}\tau_{s-p}Q_{s-p}v_s \quad (6)$$

where, for each population *p* (*p* = GrC, GoC, MLI, PC), $K_{s-p}$, $\tau_{s-p}$, $Q_{s-p}$ are the connection probabilities, synaptic decay times and quantal conductances of the connection for each presynaptic population *s* (e.g., for *p* = GrC, s = mf or GoC, hence *s-p* is either mf-GrC or GoC-GrC), $v_s$ is the presynaptic population activity in Hz computed as explained in (b).

  *b) Numerical TF.* The reference functional target to reproduce with the MF model was the neuronal spiking activity obtained in SNN simulations (*in vivo* conditions) as described in **section 2.1**. The activities of GrC, GoC, MLI and PC embedded in the SNN were simulated for different input amplitudes ($v_{drive}$) with input spikes generated from a Poisson distribution at a rate in the range 0–80 Hz, i.e. within physiological firing rate values of mossy fiber activity at rest and during tasks [66]. For each $v_{drive}$, the simulation lasted 5 seconds with a time resolution of 0.1 ms. The working frequencies of each population were extracted by averaging the spiking neuron firing rates. For each population, the outcome of the numerical TF computation was a template of dimension equal to the number of presynaptic populations, resulting in 2D numerical TFs for GrC, MLI and PC and 3D numerical TF for GoC (Fig 3A).

  The 2D numerical TF of each population was computed as the population firing rate when receiving the firing rates of the presynaptic populations, given a certain $v_{drive}$: for example, for the numerical TF template of PC, the average firing rates of MLI and GrC were computed for each $v_{drive}$ in the range 0–80 Hz. Then, these quantities were used as presynaptic signals to stimulate PC and a numerical template was obtained from the resulting PC firing rate, for each combination of presynaptic activities. The 3D TF numerical template of GoC was computed following the same strategy but considering 3 presynaptic signals (GrC, GoC, mossy fibers). GrC excitation and GoC self-inhibition were extracted from SNN simulations and the mossy fibers excitation corresponded to $v_{drive}$.

  *c) Statistical moments of the MF.* The statistical moments included in the MF are $\mu_V$, $\sigma_V$ and $\tau_V$. Starting from the conductances of the presynaptic populations, the average conductance

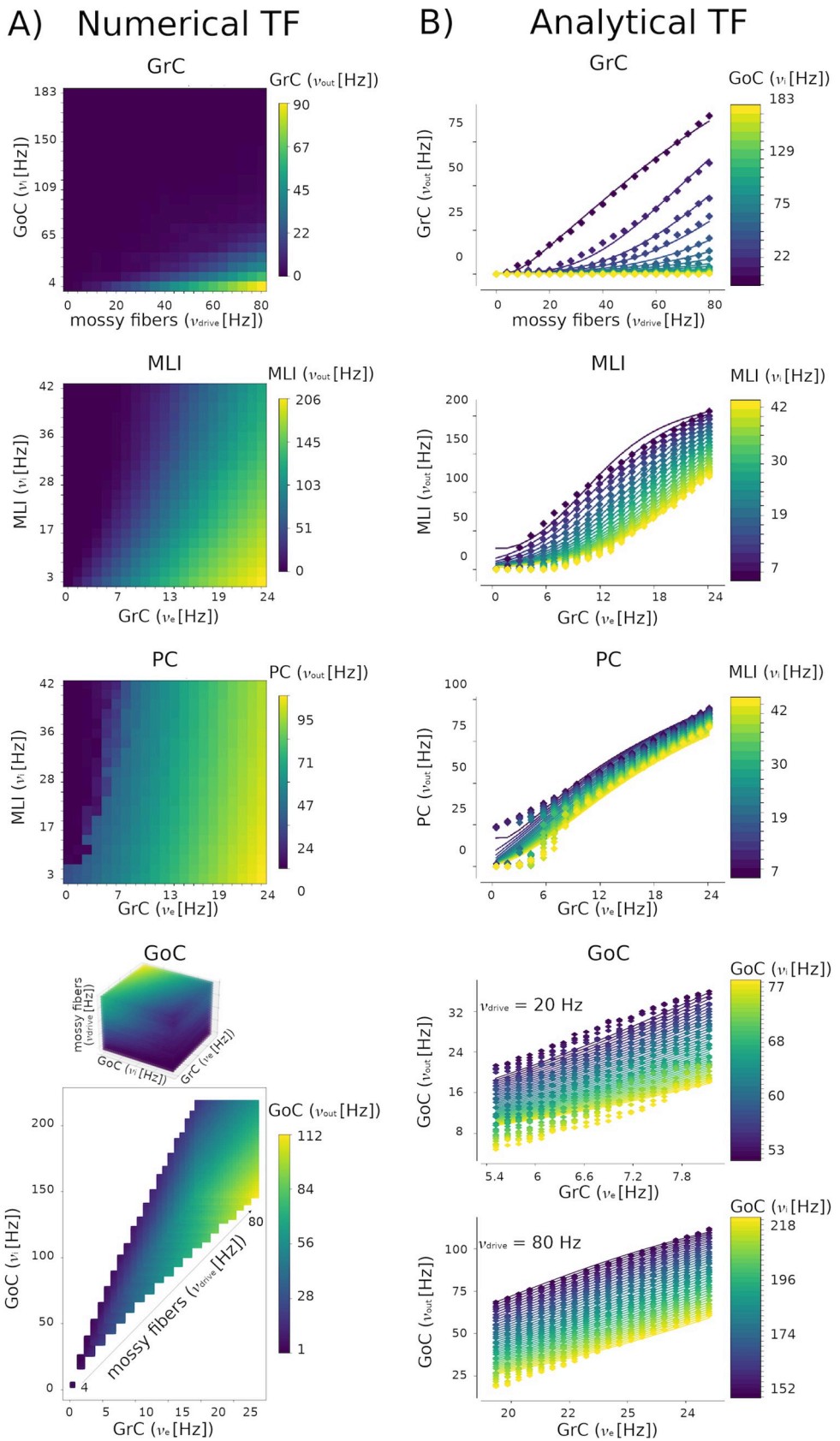

**Fig 3. Numerical Transfer Functions (TFs) and the corresponding Analytical fitting.** The simulations used to compute the numerical TFs last 5 seconds, with a time step = 0.1 ms. **A)** A 2D numerical TF template is reported for Granule Cells (GrC), Molecular Layer Interneurons (MLI) and Purkinje Cells (PC), which receive inputs from two presynaptic populations (x axis: excitatory input; y axis: inhibitory input; color bar: numerical TF color code). A 3D numerical TF template is reported for Golgi Cells (GoC), which receive input from 3 presynaptic sources, i.e., mossy fibers, GrC and GoC (showed on the axis, color bar: GoC numerical TF color code). From the 3D domain of frequencies combination, only the physiological working frequencies of GrC and GoC are considered for mossy fibers inputs from 0 Hz to 80 Hz, as obtained in corresponding spiking neural network simulations and 3D numerical TF for GoC which receive three presynaptic inputs. **B)** Numerical TFs (points) are used to fit the corresponding analytical TFs (lines, $\nu_{out}$ on y axis). Excitatory inputs are reported on the x axis, inhibitory ones are color-coded (color bar). The 2D analytical TF presents a non-linear trend, while the GoC Analytical TF shape is almost linear both for lower $\nu_{drive}$ = 20 Hz and higher $\nu_{drive}$ = 80 Hz.

$\mu_G$ of the target population reads:

$$\mu_G(\nu_s) = \sum_s \mu_{Gs} + g_L \tag{7}$$

Where $\mu_{Gs}$ is the presynaptic population conductance (Eq 4) and $g_L$ is the leak conductance of the target population (Table 1). Then, the effective membrane time constant of the target population, $\tau_m^{eff}$, is computed from $\mu_G$ as:

$$\tau_m^{eff}(\nu_s) = \frac{C_m}{\mu_G} \tag{8}$$

Where $C_m$ is the membrane capacitance (Table 1). The first statistical moment, i.e., the average of the membrane potential fluctuation, $\mu_V$, reads:

$$\mu_V(\nu_s) = e \frac{\sum_s \mu_{Gs} E_s + g_L E_L}{\mu_G} \tag{9}$$

With $E_s$ is the reversal potential of the presynaptic connection (0 mV for presynaptic excitatory populations and -80 mV for presynaptic inhibitory populations), $E_L$ is the rest potential of the target population (Table 1).

This expression is adapted from [61] to model the alpha synapses consistently with the models used in the SNN (Eqs 1, 2, and 3). Consequently, the variance and the autocorrelation

**Table 1. Fitted coefficients of the Analytical TFs.**

Phenomenological threshold.

$$V_{thre}^{eff}(\mu_V, \sigma_V, \tau_V)$$
$$= P_0 + P_{\mu V}\frac{\mu_V - \mu_V^0}{\partial \mu_V^0} + P_{\sigma V}\frac{\sigma_V - \sigma_V^0}{\partial \sigma_V^0} + P_{\tau V}\frac{\tau_V^N - \tau_V^{N0}}{\partial \tau_V^{N0}}$$
$$+ P_{\mu G}ln\left(\frac{\mu_G}{g_L}\right)$$

|  | Fitted coefficient P [V] | | | | |
|---|---|---|---|---|---|
|  | **P$_0$** | **P$_{\mu V}$** | **P$_{\sigma V}$** | **P$_{\tau V}$** | **P$_{\mu G}$** |
| **GrC** | -0.426 | 0.007 | 0.023 | 0.482 | 0.216 |
| **GoC** | -0.144 | 0.003 | 0.011 | 0.031 | 0.011 |
| **MLI** | -0.128 | -0.001 | 0.012 | -0.093 | -0.063 |
| **PC** | -0.080 | 0.009 | 0.004 | 0.006 | 0.014 |

P coefficients computed with the fitting procedure explained in [19] and extended to E-GLIF neurons

time of membrane fluctuations result in:

$$\sigma_V(v_s) = \sqrt{\sum_s (2\tau_m^{eff} + \tau_{s-p})\left(\frac{eU_{s-p}\tau_{s-p}}{2(\tau_m^{eff} + \tau_{s-p})}\right)^2 K_{s-p} v_{s-p}} \qquad (10)$$

$$\tau_V(v_s) = \frac{1}{2} \frac{\sum_s K_{p-s} v_s (eU_{p-s}\tau_{p-s})^2}{\sum_s \left((2\tau_m^{eff} + \tau_{p-s})\left(\frac{eU_{p-s}\tau_{p-s}}{2(\tau_{p-s}+\tau_m^{eff})}\right)^2 K_{p-s} v_s\right)} \qquad (11)$$

$$\text{with } U_s = \frac{Q_{s-p}}{\mu_G}(E_s - \mu_V).$$

d) *Phenomenological threshold*. The ability of the analytical template (Eq 5) to capture different firing behavior is given by the introduction of 5 parameters in the phenomelogical threshold term. The phenomenological threshold is expressed as a linear combination of the $V_m$ fluctuations properties whose coefficients are linearly fitted to the numerical TF data [19]:

$$V_{thre}^{eff}(\mu_V, \sigma_V, \tau_V) = P_0 + P_{\mu V}\frac{\mu_V - \mu_V^0}{\partial \mu_V^0} + P_{\sigma V}\frac{\sigma_V - \sigma_V^0}{\partial \sigma_V^0} + P_{\tau V}\frac{\tau_V^N - \tau_V^{N0}}{\partial \tau_V^{N0}} + P_{\mu G} ln\left(\frac{\mu_G}{g_L}\right) \qquad (12)$$

Where $\tau_V^N$ is $\tau_V$ adjusted with the ratio between membrane capacitance and leak conductance ($\frac{C_m}{g_L}$), and $\mu_V^0, \sigma_V^0, \tau_V^{N0}, \partial\mu_V^0, \partial\sigma_V^0, \partial\tau_V^{N0}$ are rescaling constants to normalize the contribution of each term [61]. P are the polynomial coefficients which are the target of the fitting procedure to compute the analytical TF as explained in e) (see Table 1).

e) *Analytical TF*. The statistical moments in Eqs 9, 10, 11 and the phenomenological threshold in Eq 12 were plugged into Eq 13 and the phenomenological threshold is computed through a fitting procedure described in [19]. Specifically, here the semi-analytical TFs are the output of two minimization algorithm with a tolerance of 10e-5: (i) Sequential Least Squares Programming algorithm and (ii) the Nelder-Mead was used to find out the optimal coefficient of the phenomenological threshold (scipy.optimize.minimize library). The TFs specific for the cerebellar populations are reported in Fig 3B.

The parameter alpha (Eq 5) was set to an optimal value for each population to fit both low and high frequencies [21]. The analytical TF, together with the statistical moments $\mu_V$, $\sigma_V$, and $\tau_V$ defined the cerebellar MF equations.

**2.2.3 Multi-layer equations.** The multi-layer MF was developed as a set of equations capturing the interdependence of the population-specific TFs, tailoring the isocortical MF described in [14] for excitatory-inhibitory networks to the cerebellar network. This formalism describes the network activity at a time resolution T which is set to ensure a Markovian dynamic of the network: T should be large enough to ensure memoryless activity (e.g., it cannot be much lower than the refractory period, which would introduce memory effects) and small enough so that each neuron fires statistically less than once per time-bin T. The choice of T is quite crucial and here it was tailored to account for cerebellar dynamics as explained in section 2.2.4.

The model describes the dynamics of the first and the second moments of the population activity for each population. The cerebellar network was built up with four interconnected populations (GrC, GoC, MLI, PC) receiving external input from mossy fibers (mf) (Fig 2), thus resulting in twenty differential equations: the four population activities ($v_{\text{GrC}}(t)$, $v_{\text{GoC}}(t)$, $v_{\text{MLI}}(t)$, $v_{\text{PC}}(t)$) and the driving input ($v_{\text{mf}}(t) = v_{\text{drive}}(t)$), the four variances of the population

activities ($c_{GrC\text{-}GrC}$(t), $c_{GoC\text{-}GoC}$(t), $c_{MLI\text{-}MLI}$(t), $c_{PC\text{-}PC}$(t)) and the one of the driving input from mossy fibers ($c_{mf\text{-}mf}$(t)), the six covariances among population activities ($c_{GrC\text{-}GoC}$(t), $c_{GrC\text{-}PC}$(t), $c_{GrC\text{-}MLI}$(t), $c_{GoC\text{-}MLI}$(t), $c_{GoC\text{-}PC}$(t), $c_{MLI\text{-}PC}$(t)) and the four covariances between population activities and the driving input ($c_{GrC\text{-}mf}$(t), $c_{GoC\text{-}mf}$(t), $c_{MLI\text{-}mf}$(t), $c_{PC\text{-}mf}$(t)). Einstein summation notation was used to report the differential system in a concise form:

$$\begin{cases} T\dfrac{dv_\mu}{dt} = (F_\mu - v_\mu) + \dfrac{1}{2}c_{\lambda\eta}\dfrac{\partial F_\mu}{\partial v_\lambda \partial v_\eta} \\[4mm] T\dfrac{dc_{\lambda\eta}}{dt} = \delta_{\lambda\eta}\dfrac{F_\lambda(\frac{1}{T} - F_\eta)}{N_\lambda} + (F_\lambda - v_\lambda)(F_\eta - v_\eta) + \dfrac{\partial F_\lambda}{\partial v_\mu}c_{\eta\mu} + \dfrac{\partial F_\eta}{\partial v_\mu}c_{\lambda\mu} - 2c_{\lambda\eta} \end{cases} \quad (13)$$

Where $v_\mu$ is the activity of population μ; $c_{\lambda\eta}$ is the (co)variance between population λ and η; $N_\lambda$ is the number of cells included in population λ. TF dependencies on the firing rate of presynaptic populations are omitted for simplicity of notation, yielding $F_\mu$ instead of $F_\mu(v_s)$ with μ = {GrC, GoC, MLI, PC} and $s$ = presynaptic population (i.e. for example, for μ = GoC, then $F_{GoC} = F_{GoC}(v_{drive}, v_{GrC}, v_{GoC})$. The model equations [13] were numerically solved using forward Euler method with an integration step of 0.1 ms. All the parameters are listed in S3 Table.

**2.2.4 Timing optimisation.** The MF time-constant T was optimised by comparing the model prediction with experimental data of the cerebellar Granular layer (Fig 4). The simulated average activity was interpolated with the experimental Local Field Potential (LFP) measured with high-density microelectrode arrays (HD-MEA) in the granular layer of acute mouse cerebellar slices [67].

LFP data were recorded at 37˚C. The external stimulus consisted in a pulse train of 5 stimuli of 50 Hz amplitude. This stimulation protocol was repeated nine times changing the HD-MEA recording channels across each experiment (Fig 4A). The LFP signals recorded were averaged across the nine experiments resulting in five values that represented the average of each pulse of the input trains. These average records were normalised on the amplitude of the signal recorded after the first stimulus [68].

The cerebellar MF simulation protocol was configured with a $v_{drive}$ = 50 Hz for 100 ms, reproducing the experimental protocol (Fig 4B). A range of plausible T values were evaluated according to literature [14,19,21,61]. MF simulations were performed with a systematic change of T values and the granular layer average activity was calculated by a weighted-mean of GrC and GoC activities. The weight of GrC and of GoC was computed according to the ratio of the spiking surfaces (GoC/GrC), which resulted in GoC/GrC = 0.13 [67]. The granular layer average activity was normalised on the maximum peak of the simulated activity, and it was interpolated with LFP recordings (Fig 4C) normalized on the maximum peak of recorded activity. For each simulation the mean absolute error between MF simulation and LFP data was computed to select the T value that minimised the discrepancy between MF and LFP signals. Since the granular layer is the driving layer of the network, the optimal T value obtained for this population was assumed to be the same for the molecular and Purkinje layers.

## 2.3 Constructive and functional validity

For constructive and *functional validity*, the cerebellar MF was tested using stimulation protocols designed to assess its ability in reproducing proper cerebellar dynamics and stimulus-response patterns.

Four different stimulation protocols were defined to reproduce the encoding of stimuli in the mossy fibers, each lasting 500 ms:

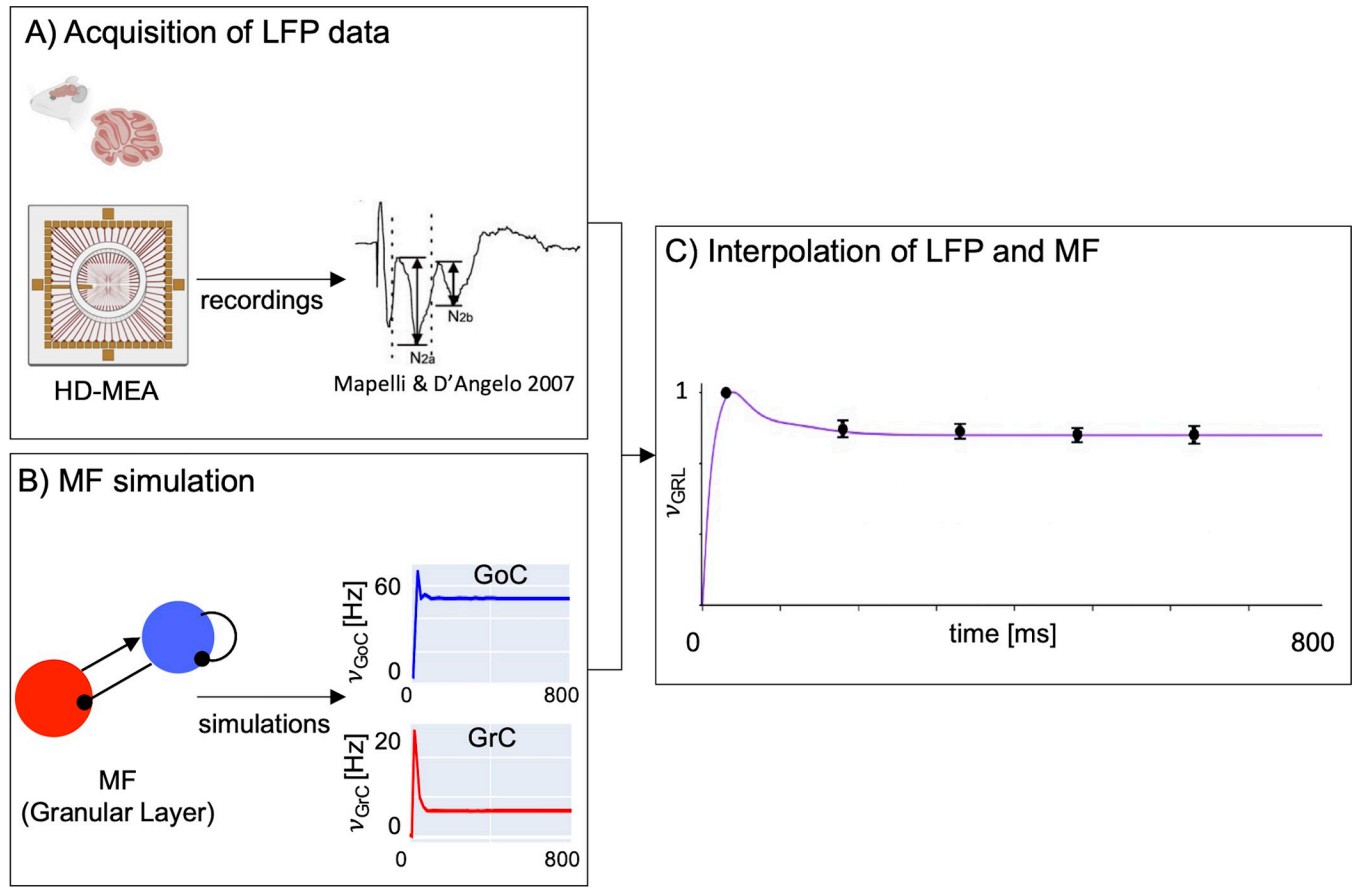

**Fig 4. The mean-field time constant. A)** Experimental acquisition of LFP. LFP signals were acquired in the cerebellar granular layer of acute mice cerebellar para-sagittal slices using HD-MEA in response to five stimulation pulse trains of 50 Hz. Figure created with Biorender.com. **B)** MF simulation. The activity of the granular layer was simulated with the cerebellar MF using a stimulation protocol emulating experimental LFP recordings. **C)** Interpolation of LFP and MF. The weighted average of the predicted Granular Layer activity($\nu_{GRL}$, violet line) interpolates the LFP data (mean ± SD; dots and bars). The relative weights of GoC and GrC are 13% and 87%, respectively. The optimal T value is 3.5 ms ± 5%, (mean ± mean absolute error between $\nu_{GRL}$ and LFP).

 i. $\nu_{drive}$ = *Step function*. A square wave with steps of amplitude 50 Hz, and lasting 250 ms to reproduce a stimulus like a sound [57].

 ii. $\nu_{drive}$ = *Sinusoidal input*. A sinusoidal input of 20 Hz amplitude and oscillation at 6 Hz to simulate the whisker movements experimental conditions [69].

 iii. $\nu_{drive}$ = *Cortical-like input*. A combination of 3 sinusoidal inputs, with fixed amplitude at 40 Hz and oscillations at 1Hz, 15Hz and 30Hz respectively reproducing an EEG-like pattern.

 iv. $\nu_{drive}$ = *Sensory-like input*. Summation of the step function described in (i) with amplitude of 20 Hz and sinusoidal function with an amplitude of 7.5 Hz and oscillating as described in (iii) to simulate a more complex input like a stimulus overlapped to a realistic basal activity.

Population activities with their standard deviations predicted by the MF were overlayed to the Peristimulus Time Histogram (PSTH) with bin = 15 ms, computed from spiking activities of the SNN in NEST simulations of the same stimulation protocols. Then, the PC activity, i.e., the output of the cerebellar cortex, was quantified as mean ± standard deviation of the firing

rate for both MF and SNN simulations. Boxplots were generated, and the Root Mean Squared Error (RMSE) was computed to quantitatively compare the output activity of MF with SNN. Correlation matrices between population activity were computed both for MF and SNN (See S1 Fig). The computational efficiency of each model was measured as computational time in seconds required for each simulation to be performed.

## 2.4 Predictive validity

For predictive validity, MF parameters were tuned to explore the MF sensitivity to modifications of local mechanisms. These modifications were derived from experimental studies on neural correlates of behavior in functional or dysfunctional conditions, focusing on inhibitory control and long term plasticity on PCs [70,71].

**2.4.1 MLI feed forward inhibition modulation.** Feedforward inhibition from molecular layer interneurons regulates adaptation of PCs. Impact of MLI-PC conductance on PC activity was explored by defining different values of MLI-PC synaptic strength $w_{MLI\text{-}PC}$ where $w_{MLI\text{-}PC}$ = 100% represents the standard condition, rates lower than 100% model disinhibited activity, while rates higher than 100% extra- inhibition.

$w_{MLI\text{-}PC}$ was added to the Analytical $TF_{PC}$ as a modulatory parameter of the presynaptic input $v_{MLI}$, resulting in a modulation of MLI contribution in PC population conductance [35] defined as follows:

$$\mu_{PC} = K_{MLI-PC}Q_{MLI-PC}\tau_{MLI-PC}v_{MLI}w_{MLI-PC} + K_{GrC-PC}Q_{GrC-PC}\tau_{GrC-PC}v_{GrC} \tag{14}$$

Each simulation lasted 500 ms with a 50Hz driving input of 250 ms after 125 ms of resting.

The Area Under Curve (AUC) of PC activity, PC peak and the depth of the pause were computed as quantitative scores for each $w_{MLI\text{-}PC}$ value AUCs and PC peaks were normalised on the respective values corresponding to the standard condition defined as $w_{MLI\text{-}PC}$ = 100%.

**2.4.2 PC Learning.** Long term potentiation and depression (LTP and LTD) are forms of synaptic plasticity at the basis of brain learning processes [72]. In the cerebellum, motor learning is driven by PC activity modulation, regulated by the plasticity of the synapses between parallel fibers (projecting from GrC) and PC, resulting in a reduction of PC activity due to LTD and in an increase of PC activity due to LTP [73,74]. To simulate the effects of GrC-PC plasticity on PC dynamics, different synaptic strengths ($w_{GrC\text{-}PC}$) were explored, including $w_{GrC\text{-}PC}$ = 65% that corresponds to the decrease of GrC-PC synaptic strength during motor learning according to animal experiment [70]. The values smaller than 100% means LTD occurred, while the values higher than 100% represent LTP occurrence. The strategy applied was analogous to equation (14).

PC AUC and PC peaks were computed for each $w_{GrC\text{-}PC}$ value as quantitative score to correlate the amount of LTD and LTP with the output activity of the cerebellar cortex.

## 2.5 Hardware and software

The SNN was built with the BSB release 3.0 (https://bsb.readthedocs.io/en/v3-last) and the numerical simulations were performed with NEST version 2.18 (https://zenodo.org/record/2605422).

The MF design, the timing optimization, the MF validation, and the MF predictive simulations were implemented in Python 3.8. Functions packages written for the present work are available on https://github.com/RobertaMLo/CRBL_MF and a DEMO is released on EBRAINS platform at https://wiki.ebrains.eu/bin/view/Collabs/mean-field-cerebellar/.

All optimisation procedures and simulations were run on a Desktop PC provided with AMD Ryzen 7 2700X CPU @ 2.16GHz with 32 GB RAM in Ubuntu 16.04.7 LTS (OS).

# 3 Results

## 3.1 The cerebellar MF

The workflow for reconstructing the cerebellar MF is shown in Fig 1 leading to a condensed representation with 4 neuronal populations for GrC, GoC, PC, MLI neurons (Fig 2). The MF was designed based on structural and functional parameters extracted from SNN simulations and the time constant was optimized with LFP experimental data. The MF working frequencies were extracted from NEST simulations of the cerebellar SNN, exploring multiple $v_{drive}$ from 4 to 80 Hz. Then these frequency ranges were used to set different plausible presynaptic signals in defining the Numerical TFs of each population (Fig 3A). The ranges were [0.42, 24.17] Hz for GrCs, [3.63, 183.15] Hz for GoCs, and [3.27, 41.66] Hz for MLIs. Note that PC working frequencies were not computed since PC activity is only projected forward to the cerebral cortex, therefore PCs never play the role of presynaptic population in this cerebellar cortical microcircuit. The α parameters that maximised the fitting performance for each population were $\alpha_{GrC} = 2$, $\alpha_{GoC} = 1.3$, $\alpha_{MLI} = 5$ and $\alpha_{PC} = 5$. The fitted coefficients P are reported in Table 1

2D Analytical TFs show a sigmoidal trend in relation with excitatory inputs (Fig 3B). GoC inhibition strongly affects the GrC Analytical TF; for $v_{GoC}$ higher than 100 Hz, GrC Analytical TF is almost 0 Hz. For low inhibition, e.g., $v_{GoC} = 13$ Hz, GrC Analytical TF is almost linear. MLI Analytical TF presents a well-defined sigmoidal trend depending on $v_{GrC}$ and modulated by the auto-inhibition, with resulting activity frequency spanning from 0 Hz up to 200 Hz. PC Analytical TF presents an increasing trend ranging from 0 to 100 Hz in relation to $v_{GrC}$ from 0 to 25 Hz, with the modulation due to the inhibitory control from MLIs. 3D Analytical TF of GoC shows a linear trend both for low and high $v_{drive}$.

The cerebellar MF resulted in a set of 20 second order differential equations including the specific population TFs, where 4 equations described the time variation of population activity, and the remaining 16 equations modelled the covariances of the interconnected populations. Fig 4 shows the result of T optimisation: for T = 3.5 ms the average granular layer activity (purple line) interpolates the experimental LFPs (red dots) with a mean absolute error of 3%.

Additionally, the overall input/output relationship of the system was analyzed by considering the input drive as a step function (see Section 2.3, protocol i) at different rates, and the output system response as the resulting PC activity. First, when input drive maintained an intensity constantly above 70 Hz, the system became unstable, and the outcome is not compatible with the physiological ranges for PC activity anymore (Fig 5A). While, within the cerebellar MF validity regime (drive in [0,70] Hz), the gain was found to be linearly dependent on the input intensity but three different sub-regimes emerged with evident discontinuities at 12 and 35 Hz (Fig 5B).

## 3.2 Constructive and functional validity

The validation of the cerebellar MF was obtained generating neuronal population dynamics with different stimulation protocols and comparing them with the corresponding SNN simulations (Fig 6). For all the protocols, the simulation lasted 500 ms and was performed with the hardware and software specified in **section 2.5.**

*Step function (i).* The MF was tested with a 250 ms@50 Hz step function on the mossy fibers, simulating a stimulus like a sound (Fig 6A) [75]. GrC activity rapidly raised at the beginning of the step input, then strongly decreased due to inhibitory GoC activity. GoCs, after an initial small peak due to both the direct incoming input and the GrC excitation, maintained a steady-state activity throughout the step duration. The dynamics of GrC-GoC interplay faithfully

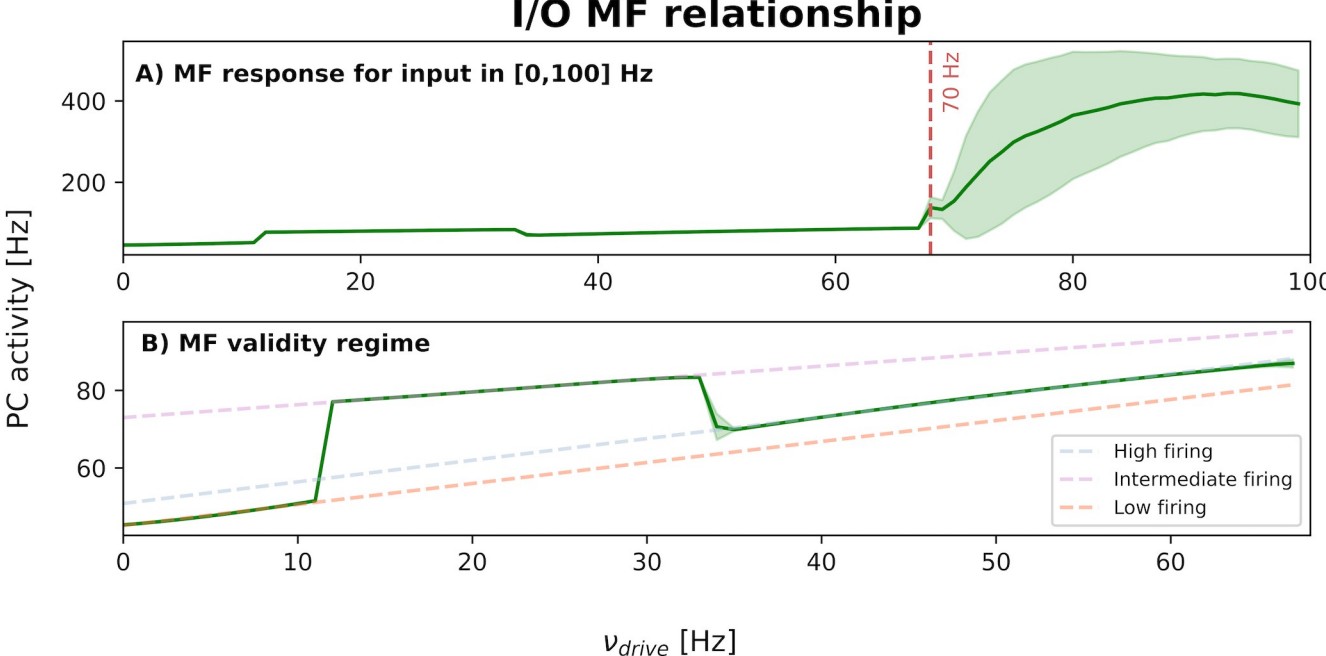

**Fig 5. Cerebellar mean-field overall input/output relationship. A)** The average response of output PC population to different levels of the driving input ($v_{drive}$) was computed, using steps at different rates in the range [0, 100] Hz as input on mossy fiber. The shaded areas show the standard deviation of PC activity computed when the $v_{drive}$ is active. As expected from the used MF formalism, the cerebellar MF works only for low input rates and specifically the critical point above which the system is not in a physiological regime anymore is at 70 Hz [48]. **B)** Within the MF validity regime, cerebellar MF I/O relationship resulted always linear, but we can identify 3 different sub-regimes depending on the input range: low firing rate in [0,12] Hz, intermediate firing in [13,35] Hz, high firing in [36,70] Hz. Each sub-regime is characterized by a different gain quantified in terms of fitting parameters obtained from a simple linear fitting procedure (f = ax+b) performed separately on each sub-regime (dashed lines).

reproduced the feedback loop between GoCs and GrCs. GrC was the excitatory input for the molecular layer, and both MLI and PC activity arose in correspondence of the GrC initial peak. Thus, exploiting network di-synaptic delays, MLIs reduced PC activity soon after its maximum, generating the typical burst-pause pattern of these neurons. The PC pause is due to both the single neuron parameters and to inhibitory local connectivity in the microcircuit. After this rapid transient, the activity of MLIs and PCs reached a steady-state. In the MF, fast dynamics at the input step onset and at the steady-state matched SNN simulations for all neuronal populations.

*Sinusoidal input (ii).* Simulated dynamics of all cerebellar populations showed oscillations driven by the input reproducing whisker movements (Fig 6B) [38]. GrC activity projected to the molecular layer a sinusoidal-shaped signal at 0.05-5Hz, contributing to an oscillatory behaviour in GoCs, and causing an oscillation in MLI between at 23–41 Hz, and in PC activity at 42–69 Hz. Oscillations had comparable amplitude in MF and SNN simulations and occurred in the same frequency ranges (except for MLI activity that was slightly higher in MF that SNN).

*Cortical-like input (iii).* Combination of sinusoids (with EEG-like frequency [76] of 1 Hz, 15 Hz, 30 Hz, respectively) caused an irregular oscillation in the input carried by mossy fibers (2–47 Hz range) (Fig 6C) Oscillations had comparable amplitude in MF and SNN simulations and occurred in the same frequency ranges.

*Sensory-like input (iv).* The summation of step function (i) with cortical-like input (iii) resulted in an irregular input (Fig 6D) depicting in-vivo noisy baseline activity with a sound

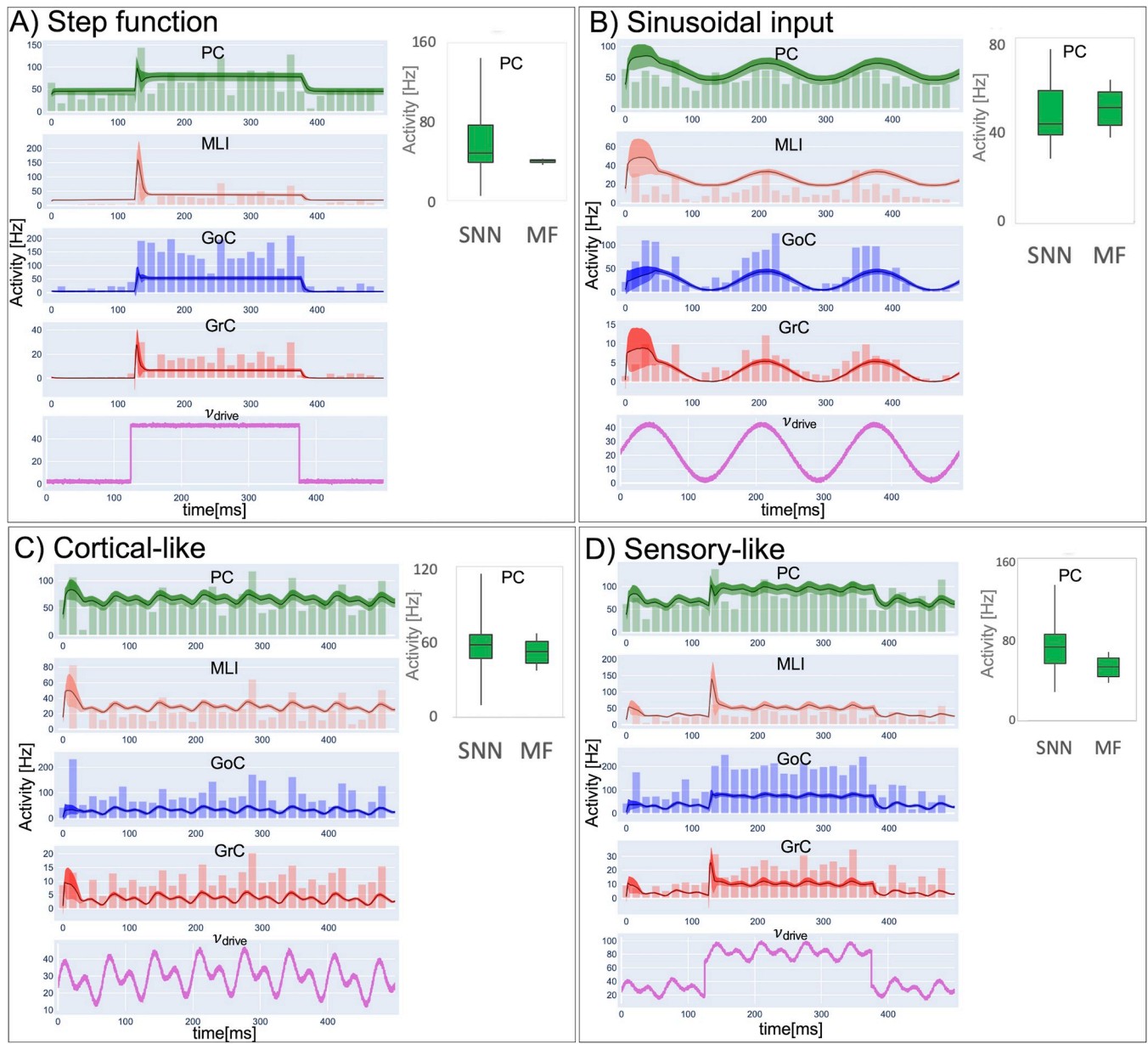

**Fig 6. Constructive and functional validity.** Comparison of Spiking Neural Network (SNN) and mean-field (MF) activity with standard deviation in cerebellar cortical populations (GrC = Granule Cells, GoC = Golgi Cells, MLI = Molecular Layer Interneurons, PC = Purkinje Cells) in response to different driving input ($\nu_{drive}$) patterns, lasting 500 ms with time resolution = 0.1 ms. **A)** step function (sound-like stimulus); **B)** sinusoidal input (whisker-like); **C)** Cortical-like input **D)** combined stimulus summing (A) step function and (C) cortical-like sinusoidal. The trace of MF activity (line = average, shade = standard deviation) is overlayed to the spiking activity, which is represented as a Peristimulus Time Histogram (PSTH, time bins of 15ms). In all cases, MF activity is within physiological ranges, capturing also fast changes of activity due to instantaneous input changes in step-function input protocols. The boxplots of PC simulated activity with SNN and MF (inset in panels), shows that the MF is able to respond to the different stimulation patterns within the same frequency ranges of SNN.

superimposed. GrC activity faithfully transmitted the driving input, with peaks at ~21 Hz. In correspondence with the GrC excitatory peak, MLIs peaked at ~130 Hz and PCs at ~100 Hz.

For each stimulation protocol, the PC activity predicted by the MF laid within the variability range of the corresponding SNN (the RMSE between MF and SNN was ~ 30%).

### 3.3 Predictive validity

The PC response to a 50Hz step-stimulus was described by a peak at 97 Hz followed by a pause down to 68 Hz; then a steady-state of 78 Hz was attained. MLI-PC and GrC-PC modulation (Fig 7) perturbed this reference condition.

**3.3.1 MLI-PC feed forward inhibition.** Inhibitory interneurons control the generation of burst-pause patterns in PC, which is fundamental for shaping the cerebellar output during motor learning [77,78]. For instance, knock-out of MLI inhibition on PC impacts on vestibulo-ocular reflex adaptation [71]. Here in MF simulations, when the MLI-PC conductance was reduced to 5% of the reference condition, the burst-pause dynamics of PC was lost, so that the PC firing settled directly back to baseline (which was elevated due to lack of inhibitory control). Conversely, when PCs were over-inhibited by the MLIs (MLI-PC conductance increased to 250% of reference condition), the pause was deeper. The PC overall activity (AUC), and PC Peak reveal an exponential trend that decays for higher MLI-PC conductances. The PC pause shows a decreasing sigmoidal trend for higher MLI-PC conductances (Fig 7A).

**3.3.2 PC plasticity.** Long Term Depression (LTD) and Long Term Potentiation (LTP) at GrC-PC synapses are thought to drive cerebellar adaptation and learning. The overall activity and the peak of PCs showed a linear and a sigmoidal trend, respectively, as the GrC-PC weight increased. With decreased GrC-PC strength, the peak was reduced or disappeared, and the steady-state activity reached lower levels. With increased GrC-PC strength, both the peak and

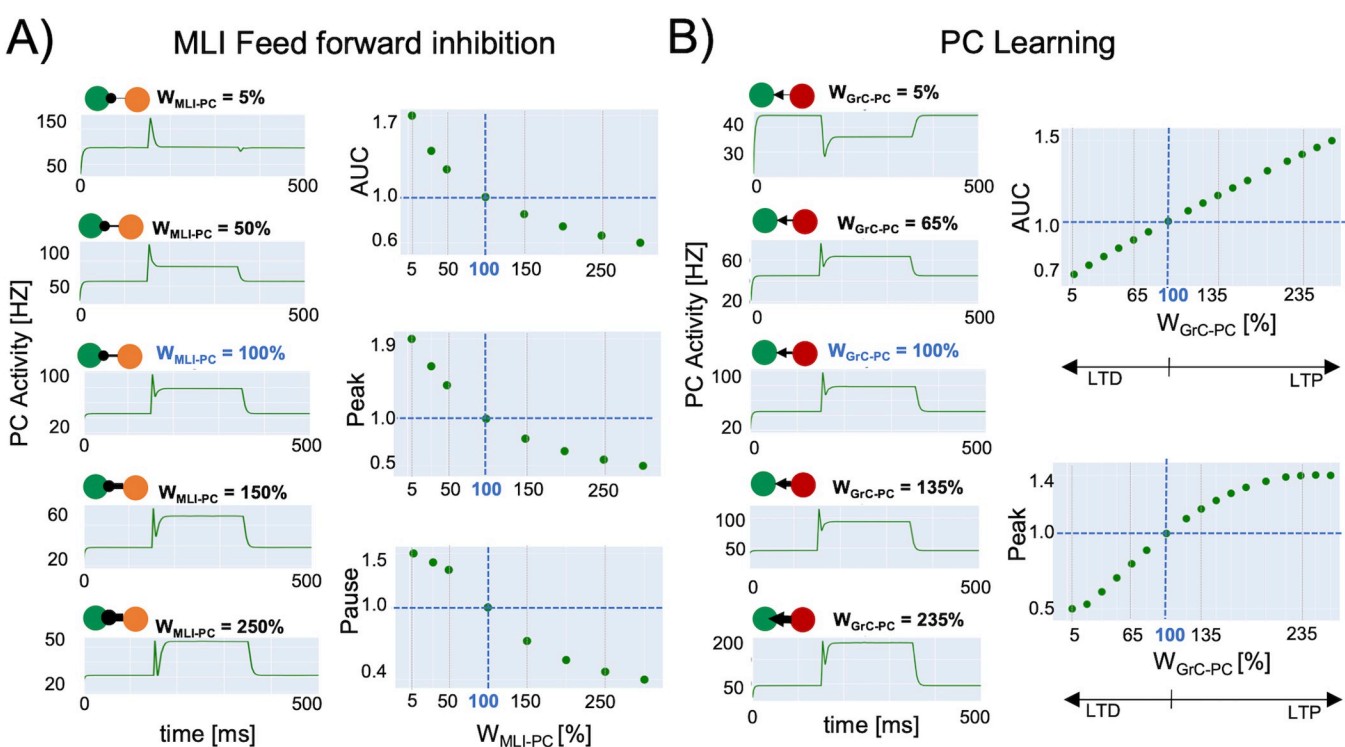

**Fig 7. Predictive validity.** Mean-field simulations using different strengths at PC connections. Simulations last 500 ms with a time resolution of 0.1 ms and *v*drive is a step at 50 Hz lasting 250 ms starting at 125 ms. Quantitative score normalized to baseline (w = 100% in blue). **A) MLI-PC feed forward inhibition**. The change of total activity (measured as Area Under Curve (AUC)) and the initial peak amplitude decrease exponentially with the MLI-PC strength. The amplitude of the pause after the peak response decreases with MLI-PC strength following an inverse sigmoidal function. **B) PC Learning** with different GrC-PC plasticity conditions (Long Term Potentiation (LTP) and Long Term Depression (LTD)). AUC linearly increases with the GrC-PC strength, matching the experimental values in experimental learning protocols. The initial peak caused by the step stimulus onset increases with GrC-PC until saturation, following a sigmoidal function.

steady-state values were increased (Fig 7B). During a typical cerebellum-driven behaviour, the eyeblink classical conditioning (EBCC), a level of suppression of about 15% has been reported and correlated with a stable generation of associative blink responses at the end of the learning process [70]. SNN simulations obtained the same result by setting AMPA-mediated parallel fibers-PC synapses = 35% [56]. In MF simulations, for $w_{GrC-PC}$ = 65%, which corresponded to a reduction of GrC-PC conductance of 35%, PC activity presented a 22% reduction of the peak and a 10% reduction of the AUC, falling into the experimental range of PC suppression [70].

## 4. Discussion

This work shows, for the first time, a MF of cerebellar cortex. According to its bottom-up nature, the MF transfers the microscopic properties of neurons (including GrC, GoC, MLI, and PC) and synapses of the cerebellar cortex into a condensed representation of neural activity through its two main statistical moments, mean and variance. The construction and validations strategies adopted here make the present MF an effective representation of the main physiological properties of a canonical cerebellar module [79].

### 4.1 MF design and validation

**The cerebellar network and TF formalism.**   The MF of the cerebellar cortex was based on the same general formalism previously developed for the MFs of the isocortex [21,80–82]. However, the cerebellar cortex MF benefitted of a previously validated cerebellar SNN to enabling to remap the multi-layer topology and the synaptic biophysical properties, avoiding collapsing these properties into an excitatory/inhibitory mono-layer network. Moreover, the input/output relationship of neurons was represented using non-linear E-GLIF models and the synapses with alpha-based conductance functions tuned on cerebellar parameters. This results in a cerebellar MF that has three key advantages over using a generic MF.

First, the parameters of population specific TFs were validated against biophysically detailed models of neurons and the connectome was derived from precise scaffold model reconstructions providing a direct link to the biological microcircuit [56].

Secondly, the equations of $\mu_V$, $\sigma_V$ and $\tau_V$ included in the TF formalism were adapted to model the alpha-shaped synapses and to maintain rise-times in synaptic dynamics. For comparison, previous MFs [19,61] used exponential synapses, which provide a less realistic approximation due to their instantaneous rise time [83].

Finally, the cerebellar MF included 4 different species of neurons that were modelled using either 2D or 3D TFs to account for the multiplicity of their inputs (Fig 3). It is worth noting that the analysis of both 2D TFs of GrC, MLI and PC, and 3D TF of GoC were fitted considering only physiological input combinations computed from single-neuron computational models. The fitting procedure allows to fine-tune the single neuron firing responses (via the TF coefficients, Eq 12) to the set of neuronal and synaptic parameters of the network. The 3D dimension of the GoC TF allowed us to keep the excitatory input from GrC and mossy fibers separate ($v_{GrC}$ and $v_{drive}$, respectively), enabling us to investigate distinct excitatory input contributions to granular layer dynamics and to the whole cerebellar MF. By fixing the excitatory mossy fibers driving input, we assessed the power of the Analytical TF in simulating spiking network activity for inputs at both low and high frequency (e.g., see Fig 3 with $v_{drive}$ = 20 Hz and $v_{drive}$ = 80 Hz).

A technical issue incurred while fitting the numerical TF. The TF formalism models the difference between the phenomenological threshold ($V_{thre}^{eff}$) and population average responses ($\mu_V$) through the complementary error function (erfc in Eq (5)), providing an immediate interpretation of how a single neuron activity was related with statistics of population dynamics

($\mu_V$, $\sigma_V$, $\tau_V$). Since the erfc is a sigmoidal-shaped function, it did not accurately follow the numerical TF distribution at the boundaries, loosing precision at high frequencies. This problem was circumvented by tuning a factor, alpha (Eq 5), similarly to what was done for isocortical models [21], which does not represent a physiological quantity, but is able to extend the TF reliably over the whole range of physiological input frequencies.

**MF tuning.** The inclusion of specific structural and functional parameters in the design of cerebellar MF (see Figs 2 and 3) generated a biology grounded model that could be validated at a mesoscopic scale with the prediction of cerebellar dynamics (multi-layer Eq 13). The dynamics of the cerebellar cortex are several times faster compared to those of the cerebral cortex [84], so that the MF time constant, T, must be optimized accordingly. The MF time constant was optimized using experimental LFP recordings from the cerebellar granular layer acquired on the same spatial scale of the MF. The best fitting was obtained by accounting for the smaller contribution of GoC than GrC activity (13% vs. 87% [85, 86]) to LFPs [67], revealing that the MF time constant of the cerebellar cortex is T = 3.5 ms with mean absolute error of 3% (Fig 3). T is definitely smaller than in cerebral cortex MFs, which can assume values up to 20 ms [21,61,81], and captures the specific high speed dynamics of the cerebellum [84]. This result further confirms the need of a MF specifically tailored on the cerebellum functional and topological parameters.

The optimal T value was plugged into Eq 13 resulting in a second order differential equation system of interdependent TFs capturing the dynamics of multiple cerebellar populations and their covariances. This rich pool of equations allowed our cerebellar MF to reproduce a variety of cerebellar dynamics in response to different inputs (see **section 2.3** and shown in Fig 5) which, by comparison with the equivalent SNN output, provided the benchmark for constructive and functional validity. A rapidly changing input like a step function reproduced a stimulus [75] carried by the peripherical mossy fibers, causing rich dynamics in the cerebellar cortex including the typical PC burst-pause responses [87]. Adding a multi-sinusoidal input to the stimulus replicated a more physiological condition accounting for background activity. This resulted in rich PC dynamics, which still maintained burst-pause responses. A sinusoidal input was meant to emulate more complex experimental conditions, like those determined by whisker movement [38,50,88–91]. In particular, a multi-sinusoidal waveform was used to emulate composite inputs from the cerebral cortex [76,92]. This makes the MF model of the cerebellum an appropriate tool to investigate different pathological conditions, that could eventually be tested in the context of a cerebro-cerebellar loop.

The cerebellar MF reproduced the known aspects of circuit physiology revealed experimentally. These include: the ability of GrCs both to respond to impulsive inputs with bursts curtailed by GoC inhibition and to faithfully follow slow input fluctuations [93]; the ability of PCs to generate burst-pause responses accentuated by MLIs [48,51]. The cerebellar MF behaved as a smoothing filter reproducing the trend of SNN simulations without capturing the intra-population variability (e.g., GrC fluctuations in response to a constant input—Fig 6A). Since the intra-population variability was not accounted for, the maximal GoC responses were underestimated by the MF, which approximated GoG to a homogeneous population neglecting their biological heterogeneity [87]. The differences that could be observed between the MF and the corresponding SNN simulations could thus be explained as a drawback of the homogeneity assumption. Despite these aspects (further discussed in section 4.4) the MF predicted with good approximation the SNN cerebellar output, i.e., the PC activity (Boxplot in Fig 6).

## 4.2 MF predictions

A critical step in model validation is to demonstrate its ability to predict functional states not used for model construction. The cerebellum is well known for the ability to change its

network dynamics because of synaptic and non-synaptic plasticity. However, differently from SNN, the MF did not include plasticity mechanisms. Thus, we directly tested the MF ability to predict the effects of plasticity expression by mapping a set of precomputed synaptic changes on the MF itself. MF reproduced the impact of changes in MLI-PC conductance confirming that it can reproduce the complex burst-pause behaviour of PC, tuned through the MLI-PC connectivity.

The cerebellar MF reproduced the experimental recordings in EBCC experiments on behaving mice [70] pointed out a PC LTD of 10% in terms of overall activity and 22% for the peak (Fig 7 $w_{GrC-PC}$ = 65%). This protocol corresponds to a reduction of 35% of AMPA-mediated parallel fibers-PC in the corresponding SNN simulation, therefore our MF capability of capturing synaptic mechanism is further validated against in-vivo recordings.

In aggregate, the cerebellar MF was able to reproduce complex physiological mechanisms and predict the activity changes caused by synaptic modulation [56,70]. Although the cerebellar MF was able to replicate changes in activity patterns characterizing learning states, it was not possible yet to simulate the long-term plasticity rules that would be needed to autonomously generate the PC learning curves. Therefore, this MF is a flexible tool that can be used to investigate physiological or pathological properties by tuning the input of the TF on a target population ($TF_{PC}$ in these simulations), without complicating neither the TF fitting procedure nor the model equations. In contrast to conventional MFs, which are often a black box in terms of physiological parametrization, the cerebellar MF internal model parameters inform about the average properties and variance of fundamental mechanisms in the circuit (e.g., intrinsic and synaptic excitation) and can therefore be used to remap biological properties.

The procedure of parameters tuning might pave the way for further manipulation to remap physiological and/or pathological features onto the MF. As an example, identifying and extracting biophysical meaningful features from atlases, such as the receptor maps (e.g., extracted form Allen Brain Atlas, http://ww.brain-map.org), might allow combining mesoscopic MF simulation with characteristics of molecular functions.

## 4.3 Performance vs. realism

The MF approximated a complex SNN of ~$3x10^4$ neurons and ~$1x10^6$ synapses (**section 2.1**) with 20 equations reducing the computational time by 60% with lower memory requirements. Nevertheless, TFs fitting could be improved replacing the procedure in **section 2.2.2** with a lookup table-based algorithm, which might yield to a gain in computational time up to an order of magnitude. This will represent a definite advantage when performing long-lasting simulations reflecting the acquisition time of *in vivo* recordings like EEG and fMRI or when simulating learning processes in closed-loop controllers. On the other hand, thanks to its bottom-up nature it was possible to maintain the biological realism in responses to various stimuli, including simulations of learning-induced firing rate changes and pathological conditions at the neuronal population level, obtaining a good balance between computational load and biological plausibility. This will allow to make predictions on the underlying neural bases of ensemble brain signals and to identify the elementary causes of signal alterations in pathology.

Theoretically, the cerebellar MF is not just a replacement for SNN, but rather a specific instrument for addressing questions at a mesoscale. SNNs and MFs indeed cover complementary aspects of brain function and dysfunction. For example, in Parkinson disease, a SNN was successfully used to simulate the thalamic and basal ganglia microcircuits in order to control deep brain stimulation electrodes [94,95]. The SNN provided a description of microcircuit activity and of the involvement of dopaminergic neurons in tremor [96,97]. On the other

hand, MFs of the basal ganglia and thalamocortical circuit were used to embed neural mechanisms into the motor network involved in Parkinson disease and investigate large-scale oscillations [98]. In dystonia, a cerebellar SNN embedding Dystonic alterations was successfully used to simulate alterations in PC firing rate, olivocerebellar connectivity, and parallel-fiber/PC spike-timing dependent plasticity (STDP) [99]. While these microscopic properties may not be reproduced by the cerebellar MF, it may however reproduce the overall PC output alterations allowing to simulate the large-scale propagation of cerebellar abnormalities to other brain regions.

### 4.4 Limits and further developments

The MF presented here simplifies the representation of neural activity and enables computationally efficient analysis of network dynamics. The tradeoff of simplification is the loss of some properties that the real system has. In general, MFs present a coarser granularity than SNNs, limiting studies in which single neurons, single spikes, or neural heterogeneity matters. Furthermore, MFs and SNNs perform a different information encoding. SNNs encode information with spike-time precision, while MFs are rate-coded machines modeling the average activity of entire neuronal populations. For example, STDP may not be simulated with a MF, where single spike times are not represented.

The chosen mathematical formalism can also impact on the performance of the MF. MFs based on stationary TFs are not intended to process frequency oscillations unlike MFs resulting from analytical derivations of the time-varyng rate (e.g. available for two populations model) [100,101]. Specifically, translating the cerebellar SNN into the cerebellar MF involves simplifications (e.g., lack of plasticity rules) and assumptions (e.g., homogeneity inside the populations). In practice, this results in a cerebellar MF with the predictive capability of reproducing the effects of complex dynamics like burst-pause of PCs at a population level, but without modelling the neuronal coding over an heterogeneous neuronal population [102,103]. For example, a cerebellar SNN was used to simulate the control of saccades movements exerted by two subpopulation of PCs [104]. Conversely, the cerebellar MF assumes intra-population homogeneity and can't be used to study such phenomena.

Besides the limitations intrinsic to a population-level formalism, the cerebellar MF could be further developed by adding climbing fibers (cfs) as an additional input to PCs. The PC TFs including cfs input could catch further complexity and MF predictions might generate burst-pause responses. Another improvement of the cerebellar MF could include synaptic plasticity rules like in the corresponding SNN [105]. Such mechanisms at a mesoscopic scale could be modeled by implementing changes of synaptic strength based on the firing rates.

Considering the regime of applicability, the formalism of our cerebellar MF is intended to work at an asynchronous state with irregular firing rate. Therefore, input oscillations are not processed by this formalism [106,107]. Oscillations in Fig 6 are merely input-driven. To include frequency elaborations as intrinsic processes of the cerebellar MF itself, further improvements in the MF formalism should be introduced, e.g., embedding mechanisms to account for adaptation and neuronal heterogeneity [108]. The proposed improvements of the cerebellar MF presented here are all possible in principle and their implementation will depend on the specific use of the MF in future applications.

### 4.5 Conclusions and perspectives

In aggregate, the cerebellar cortex MF enforces a bottom-up approach tailored to the specific structural and functional interactions of the local neuronal populations and has a robust constructive and functional validity. By accounting for a variety of representative patterns of

discharge in cerebellar cortical neurons, the present MF can be considered a proxy of the real biological network. The cerebellar MF model developed here was validated against mice data and our promising results prompt for its translation to humans. In principle, the fundamental network architecture will remain the same, while parameters of neuron and synaptic models will require specific retuning. Further model validations should be implemented on histological and electrophysiological data from human cerebellar slices. The internal model parameters inform about the average properties and variance of fundamental mechanisms in the circuit, namely intrinsic and synaptic excitation, and can therefore be used to remap biological properties onto the MF [28]. In future applications, this will allow to tune the MF towards specific functional or dysfunctional states that affect the cerebellum. Among these it is worth mentioning ataxias [109,110], paroxysmal dyskinesia [111,112], dystonia [113,114], autistic spectrum disorders [115,116] as well as other neurological pathologies like multiple sclerosis [117–119], dementia [39] and Parkinson disease [120,121], in which a cerebellum involvement has been reported. The cerebellar MF could be applied to whole-brain simulators using TVB, similarly to what has been done already for the isocortical MF [14,122–125]. Different modules of cerebellar MFs could be connected one to another and with other cortical and subcortical MFs following subject-specific connectome rules that could be extracted from imaging techniques such as connectivity matrices computed from diffusion weighted imaging [126]. Considering the specificity of signal processing in different brain regions, this approach represents a leap ahead towards adopting specific neural mass models for different brain regions.

On the theoretical side, TVB simulations using classical neural masses [127] for all brain nodes [39,40] can now be compared to those using the cerebellum MF. At the other extreme of the spectrum, TVB with embedded cerebellar MF can be compared to TVB-NEST co-simulations [95], in which spiking neurons are represented explicitly [52,54,56,57].

In conclusion, the cerebellar MF represents the first step toward a new generation of models capable of bearing biological properties into virtual brains that will allow to simulate the healthy and pathological brain towards the overarching aim of a personalized brain representation and the technology of brain digital twins [128].

## Supporting information

**S1 Table. Neuron parameters.** Parameters specific of the type of neurons included in the multi-layer MF populations. The parameters in the top part are chosen according to literature, while the parameters at the bottom were extracted from spiking neural network simulating the cerebellar cortex spiking activity. mf = mossy fibers, GrC = Granule Cells, GoC = Golgi Cells, MLI = Molecular Layer Interneurons (Basket cells and Stellate cells)
(DOCX)

**S2 Table. Presynaptic parameters.** Parameters used to set up the inter-population connectivity of the multi-layer MF cerebellar network. Parameters were extracted from the spiking neural network simulating the cerebellar cortex spiking activity. mf = mossy fibers, GrC = Granule Cells, GoC = Golgi Cells, MLI = Molecular Layer Interneurons (Basket cells and Stellate cells). K = pre-synaptic connectivity resulting by weighting the mean synaptic convergence with the number of synapses; Q = quantal synaptic conductance in nS; $\tau$ = synaptic decay time constant; $E_{rev}$ = reversal potential that is 0 V for excitatory synaptic connections and -80 V for inhibitory synaptic connections.
(DOCX)

**S3 Table. Mean-field symbols.**
(DOCX)

**S1 Fig. Correlation matrices between population activities, in MF and SNN networks.** Population activities predicted by cerebellar MF were highly cross-correlated, resulting in Pearson Correlation Coefficients always $> 0.7$ (Panel A). Correlation between population activities simulated by SNN is reported in Panel B; variability intrinsic in SNN led to lower correlations between pairs of population activities than in MF. These correlations matrices are reported for each input pattern. The input frequencies might be replaced with a probabilistic kernel to introduce variability in the MF formalism and reduce the inter-population correlations. (TIFF)

## Acknowledgments

We thank Robin De Schepper for useful discussion on Brain Scaffold Builder framework and Alessio Marta for an insight into the mathematical formalism of mean field.

## Author Contributions

**Conceptualization:** Claudia A. M. Gandini Wheeler-Kingshott, Fulvia Palesi, Claudia Casellato, Egidio D'Angelo.

**Data curation:** Roberta Maria Lorenzi.

**Formal analysis:** Roberta Maria Lorenzi, Marialaura De Grazia.

**Funding acquisition:** Yann Zerlaut, Alain Destexhe, Claudia A. M. Gandini Wheeler-Kingshott, Fulvia Palesi, Egidio D'Angelo.

**Methodology:** Roberta Maria Lorenzi, Alice Geminiani.

**Project administration:** Claudia Casellato, Egidio D'Angelo.

**Supervision:** Yann Zerlaut, Alain Destexhe, Fulvia Palesi, Claudia Casellato, Egidio D'Angelo.

**Validation:** Roberta Maria Lorenzi.

**Visualization:** Roberta Maria Lorenzi.

**Writing – original draft:** Roberta Maria Lorenzi, Alice Geminiani, Egidio D'Angelo.

**Writing – review & editing:** Roberta Maria Lorenzi, Alice Geminiani, Yann Zerlaut, Marialaura De Grazia, Alain Destexhe, Claudia A. M. Gandini Wheeler-Kingshott, Fulvia Palesi, Claudia Casellato, Egidio D'Angelo.

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
