## [Decision Letter · Decision Letter 0]

6 Apr 2023

Dear Dr. Lorenzi,

Thank you very much for submitting your manuscript "A multi-layer mean-field model of the cerebellum embedding microstructure and population-specific dynamics" for consideration at PLOS Computational Biology.

As with all papers reviewed by the journal, your manuscript was reviewed by members of the editorial board and by several independent reviewers. In light of the reviews (below this email), we would like to invite the resubmission of a significantly-revised version that takes into account the reviewers' comments.

Three top experts reviewed your work and they had major concerns in particular about the match between the model and experimental evidence. We would ask you to only submit a revised version if you can address these concerns (in particular those by Reviewer 1 and 3) with an updated model. The work is novel and interesting for the field but the expectations from the field do not seem to be satisfied in the current version.

We cannot make any decision about publication until we have seen the revised manuscript and your response to the reviewers' comments. Your revised manuscript is also likely to be sent to reviewers for further evaluation.

Sincerely,

Hermann Cuntz

Academic Editor

PLOS Computational Biology

Daniele Marinazzo

Section Editor

PLOS Computational Biology

Three top experts reviewed your work and they had major concerns in particular about the match between the model and experimental evidence. We would ask you to only submit a revised version if you can address these concerns (in particular those by Reviewer 1 and 3) with an updated model. The work is novel and interesting for the field but the expectations from the field do not seem to be satisfied in the current version.

Reviewer's Responses to Questions

**Comments to the Authors:**

Reviewer #1: This manuscript translates an highly detailed SNN model of the cerebellum into a significantly simpler mean-field model. The manuscript is very relevant and the research therein has great value. According to the authors this is the first time this is done for the cerebellum, which is correct to my knowledge. The many ways in which biology underlies the SNN model are very compelling, particularly microanatomy, biophysics, and the translation into an MF model is valuable from the context of whole brain models. A MF model of the cerebellum has great value and potential in explaining mesoscopic behavior of electric fields and behavior of populations under a variety of conditions.

However, the shortcomings of a model are as interesting as its power and merit in -depth analysis of the deviations between the SNN and MF. Rather than repeatedly bolstering the value of the type of modeling, the authors should focus on systematically evaluate the cases in which the model is applicable. There are a few sentences that aim at this objective, but IMHO they often offer shallow reasoning. Similarly, both introduction and discussion lack depth.

It seems to me that the manuscript finds it difficult to find a balance between the simplifications necessary for the MF model and its explanatory power. There are multiple statements trying to justify its use and exagerating its capabilities. At times it seems to me that the authors are not fully compelled by their own model and hope that by repeating the claims, these are justified. Perhaps if the manuscript is improved to cover some of those use cases, the authors would not need to go to such lengths to argue their approach. In fact, it does not appear to me that MF models need to be justified at all, they have already shown their value in a large number of publications over the last 40 years. The manuscript should maintain a more critical and factual stance, focusing on objective description and curtailing the sales pitch. Authors should prefer a neutral language throughout the manuscript.

For instance, the manuscript belabors the point that the MF model is of great biophysical relevance (e.g. section 4.3), but the discussions remains abstract and do not sufficiently establish the scope of use beyond multiple mentions to whole brain modeling / pathology. Clearly some pathologies are betters studied via spiking models. Please discuss with more clarity: When should we consider an MF rather than a spiking model?

In terms of scope and limitations multiple aspects of the translation from SNN to MF merit deeper discussion. Clearly, in terms of explaining cerebellar computations, a continuous system such as this fails to capture many relevant aspects of cerebellar operation. For example:

- CF related synchronous pauses in PN are impossible

- Influence of local microcircuit topology AND topography (e.g. directionality of PN collaterals)

- Can it be used to represent the surfaces of the neuropils (can it become a PDE)?

- Does low dimensional representation of spikes preclude representation of high dimensional sensory inputs?

- How to study reorganization at cell level?

- What about coupling at different temporal and spatial scales (gap junctions / glutamate spillover)

- What about the significantly reduced variability of responses in comparison to the SNN

- Does the size of the population being represented influences the time constants of the model?

- How about oscillations in the intrinsically active SNN populations? Are there any, or can we exclude this from the model?

Other Comments:

The listed motivation for developing the MF model is that a brain region specific MF is better than a generic, existing MF because these “[...] do not capture in full the complex properties of specific neuronal populations [...]”. This is never quantified, and we never see an actual comparison to such a generic MF model - leaving the question, does it actually perform better than the generic model? The justification to develop the model is purely based on the availability of data (line 106). Yet, in the discussion, it is stated that “However, the cerebellar cortex MF benefitted of a previously validated SNN [..]” - this statement is not backed up and I’m not yet convinced that this is an improvement over a generic MF model (I mean personally I might be convinced).

Chosen model complexity: A semi-analytical TF is used to define the MF. This TF is started as a template. Where did this specific template come from? Why aren’t there more/less terms in the model template (seems like the alpha term is explained partly in line 600 in the discussion)? Were other models considered?

Limitations (discussion): Among other simplifications, only the 4 main neural populations of the cortex were modeled; how does this show up in the results? In line 636 GoC behavior is discussed, is this the only behavior does this model not capture correctly? In what cases would one be better off using the more detailed SNN?

Underlying model: Why is the chosen cerebellar cortex SNN the right underlying model to develop the MF model? No comparison to other cerebellar cortex models is made (do these exist?), and why then this exact model is chosen.

Figure 3, Multiple readability issues. Also, in B due to overlapping color schemes the overlap between lines and points is confusing. Legend needs edits for readability.

Figure 5): “[..] the MF is able to respond to the same different stimulation patterns within the same frequency ranges of SNN” -> 1) This is never quantified (should I base this statement on the boxplots being next to each other?). 2) For the step function I wouldn’t necessarily agree that the frequency ranges overlap. What creates the differences here? This deserves careful discussion as it is one of the main differences between SNNs and MF and define the scope of applicability of the MF model.

Nitpicks:

Line 164, equation (1), page 8. Missing brace in the first equation.

Line 230, equation (5) use \\mathrm{erfc} or \\operator{erfc}

Line 246, equation (6) extraneous dot after equals sign

Line 292, equation (10) missing bracket under the fraction

Line 297, firing ‘behaviors’

Line 332, equation (13) why is there a large bracket on the right side?

Line 335, repeating “Einstein’s notation”. Also, IMO this adds unnecessary opacity, as this is not known to neurobiologists to whom this paper is addressed.

Line 409, listing the synaptic strengths is absolutely not necessary

Line 413, ‘smaller than’ rather than ‘minor’

Line 415, ‘peaks’

Line 286, equation (Es=) no space? Why an = sign in a sentence (also later in the sentence)

Figure 4B): missing X axis (time?)

Line 572: remove “precisely”. Also, attempt to maintain a more neutral language throughout the manuscript.

Line 587, remove “indeed”

Line 592: =vdrive=

Line 598-599 need editing for clarity. What does it mean that the “erfc” is stiff?

Line 603: remove “precise” as those are in the SNN and not in the MF. An approximation of a precise description is still an approximation.

Line 605: instead of ‘higher scale’, write at a ‘mesoscopic scale’ or ‘coarser’ scale.

Line 613: AFAICS, this argument does not clearly follow from the preceding.

Line 628: whether it is a ‘powerful tool’ is yet to be shown. In the absence of that demonstration, this sentence is void.

Line 643: consider ‘dynamics’ instead of ‘functioning’

Reviewer #2: Uploaded as an attachment.

Reviewer #3: Lorenzi et al. present a study on the simulation of the cerebellar cortical neural circuit using the mean-field (MF) framework. While the MF method has potential for studying the cerebellar neural circuit, the authors have not made a strong case for publication. Overall, my rating of this paper is not favorable for publication or further review due to the lack of rigorous comparison between the reference and approximation, inconsistency with in vivo experimental data, and lack of testing with realistic stimuli.

My first major concern is the lack of rigorous and quantitative comparison between the reference result (SNN) and the approximation. For example, in Figure 3, the authors show transfer functions of neurons, but there is no quantification, only the best-fit parameters are shown. Similarly, in Figure 5, there is no quantification of the resemblance between SNN and MF simulation results. The authors claim similarity, but the blue curve undershoots the GoC PSTH, the orange curve overlays on top of the MLI PSTH, and the red curve does not capture the temporal dynamics of the GrC firing. On the right hand side, box plots the mean firing rate (?) is shown, but again no statistical comparison either.

Second, the simulation results are inconsistent with in vivo experimental data. For instance, the modulation of the firing of the mossy fibers by various oscillations does not entrain the Purkinje cells, as shown in Ros et al., J Neurosci 2009. Additionally, the authors explore different synaptic parameters in Purkinje cells in Figure 6, but do not show heterogeneous responses that the Purkinje cells show potentially due to different excitation/inhibition balance in their inputs (Jelitai et al., Nat Comm, 2015). Particularly, the simulated Purkinje neurons never show the pausing response except for a unrealistic corner of their parameter space (Fig. 6B Top). This casts doubt on the biological relevance of their results.

Lastly, the MF simulation is tested only with limited or unrealistic protocols for external stimuli. For example, the authors did not test their models with mossy fiber bursts, which are commonly observed in vivo. Mossy fibers burst with the peak firing rate ~500 Hz with sensory stimuli (Rancz et al., Nature, 2007) and the similar phenomena are reported for the mossy fibers carrying motor information (Ohtsuka and Noda, Neurosci Res, 1992). The absence of this protocol limits the applicability of this study to real biological phenomena.

**Have the authors made all data and (if applicable) computational code underlying the findings in their manuscript fully available?**

Reviewer #1: Yes

Reviewer #2: **No: **The linked repository is not public: https://github.com/RobertaMLo/CRBL_MF_Model

Reviewer #3: **No: **The paper mentions a github repo, but it returns 404.

PLOS authors have the option to publish the peer review history of their article (what does this mean?). If published, this will include your full peer review and any attached files.

Reviewer #1: No

Reviewer #2: **Yes: **Jan Fousek

Reviewer #3: No
---

## [Decision Letter · Decision Letter 1]

15 Aug 2023

Dear PhD Lorenzi,

We are pleased to inform you that your manuscript 'A multi-layer mean-field model of the cerebellum embedding microstructure and population-specific dynamics' has been provisionally accepted for publication in PLOS Computational Biology.

Best regards,

Hermann Cuntz

Academic Editor

PLOS Computational Biology

Daniele Marinazzo

Section Editor

PLOS Computational Biology

Reviewer's Responses to Questions

**Comments to the Authors:**

Reviewer #1: The paper presents a valuable contribution to the cerebellar literature and presents a method to produce mean field models for the cerebellar circuit, potentially useful in a variety of scenarios.

Reviewer #2: The authors have addressed my comments. I only noticed following minor mistake: reference 13 (Fellner et al 2022) is not a relevant reference for finite effects in the mean field models as stated in the manuscript, but rather deals with FEM modeling of stimulus.

**Have the authors made all data and (if applicable) computational code underlying the findings in their manuscript fully available?**

Reviewer #1: Yes

Reviewer #2: **No: **

PLOS authors have the option to publish the peer review history of their article (what does this mean?). If published, this will include your full peer review and any attached files.

Reviewer #1: No

Reviewer #2: **Yes: **Jan Fousek

---

## [Editor Report · Acceptance letter]

23 Aug 2023

PCOMPBIOL-D-22-01885R1 

A multi-layer mean-field model of the cerebellum embedding microstructure and population-specific dynamics

Dear Dr Lorenzi,

I am pleased to inform you that your manuscript has been formally accepted for publication in PLOS Computational Biology. Your manuscript is now with our production department and you will be notified of the publication date in due course.

With kind regards,

Zsofia Freund
